# The Patient Centered Assessment Method (PCAM) for Action-Based Biopsychosocial Evaluation of Patient Needs: Validation and Perceived Value of the Dutch Translation

**DOI:** 10.3390/ijerph182211785

**Published:** 2021-11-10

**Authors:** Rowan G. M. Smeets, Dorijn F. L. Hertroijs, Mariëlle E. A. L. Kroese, Niels Hameleers, Dirk Ruwaard, Arianne M. J. Elissen

**Affiliations:** Department of Health Services Research, Faculty of Health, Medicine and Life Sciences, Care and Public Health Research Institute (CAPHRI), Maastricht University, 6200 MD Maastricht, The Netherlands; d.hertroijs@maastrichtuniversity.nl (D.F.L.H.); marielle.kroese@maastrichtuniversity.nl (M.E.A.L.K.); niels.hameleers@maastrichtuniversity.nl (N.H.); d.ruwaard@maastrichtuniversity.nl (D.R.); a.elissen@maastrichtuniversity.nl (A.M.J.E.)

**Keywords:** person-centered care, primary health care, biopsychosocial model of illness, chronic disease, psychometrics, mixed-methods

## Abstract

The Patient Centered Assessment Method (PCAM) is an action-based tool that supports professionals to engage in a biopsychosocial assessment with patients and measure their needs. It is a promising tool for person-centered care. As the Netherlands lacks such a tool, a Dutch version was developed. Furthermore, we aimed to contribute to the relatively limited insights into the psychometric properties and value of the tool when used as part of a needs assessment in primary care. Confirmatory factor analysis was used to study construct validity and Cronbach’s alpha was computed to assess reliability. Furthermore, we interviewed 15 primary care professionals who used the PCAM. It was confirmed that each PCAM domain measures a separate construct, informed by the biopsychosocial model. The tool showed adequate reliability (Cronbach’s alpha = 0.83). Despite face validity concerns, the tool was mainly valued for measurement of patient needs and to facilitate action planning. Criticism of the PCAM pertained to a limited focus on the patient perspective, which is one of the crucial aspects of person-centered care. These rich, mixed-method insights can help to improve the value of the PCAM, as one of the few multifunctional tools to support professionals in holistic assessments.

## 1. Introduction

The Patient Centered Assessment Method (PCAM) was developed by Maxwell and colleagues in 2013 to support holistic assessment of biopsychosocial patient needs in primary care [1,2]. The PCAM includes 12 items clustered into four domains, i.e., health and well-being; social environment; health literacy and communication; and service coordination. Each item is scored using a four-point traffic light-style system indicating the growing need for (professional) action, ranging from ‘routine care’, ‘active monitoring’, and ‘plan action’ to ‘act now’ [1]. Hence, although the PCAM is primarily a conversation tool to take a comprehensive, person-centered approach to patients, it also supports measurement and monitoring of patient needs [1]. To make a shared decision about the required ‘actions’ (e.g., referral, behavior change intervention) for a patient, also called action planning, the tool ends with four questions. These relate to what action is needed, who needs to be involved, what barriers exist, and what action will be taken.

Although the PCAM was originally designed for primary care, insights into the feasibility and perceived value of applying the tool in this setting are relatively scarce. A substantial part of the existing studies of the instrument have been conducted in the context of transitional or hospital care [3,4,5,6]. However, available primary care studies conclude that PCAM is a feasible and valuable tool that supports holistic assessment and allows for referral to a spectrum of services [1,2,7]. Insights into the psychometric properties of the tool are also relatively sparse and ambiguous. Although existing studies conclude that the PCAM has good internal consistency, insights into the theoretical constructs (also described as ‘factors’) measured by the tool are conflicting [1,5]. Maxwell et al. [1] studied a former version of PCAM and concluded that the domains ‘health and well-being’, ‘social environment’ and ‘health literacy and communication’ each constitute a separate theoretical construct, followed by one question related to required actions. In contrast, Yoshida et al. [5] distinguished two constructs underlying the current 12-items PCAM tool. These were ‘patient-oriented complexity’, related to internal health determinants (e.g., health literacy), and ‘medicine-oriented complexity’, related to the external health determinants (e.g., service coordination).

Professionals need support to engage in holistic conversations with patients, but a valid, reliable, and feasible tool that is sensitive to the biopsychosocial needs of patients is still missing in the Netherlands [8,9]. Therefore, we aimed to: (1) translate and contextualize the PCAM for use in Dutch primary care. Furthermore, as there is a knowledge gap concerning the psychometric properties and value of the PCAM in primary care, we formulated two additional research aims relevant for an international context; (2) to increase insight into the psychometric properties, i.e., the (construct) validity and reliability of the tool, by testing and comparing both previously identified factor structures to determine the best-fitting structure [1,5]; and (3) to assess the perceived value, feasibility and face validity of the PCAM when used to support a person-centered needs assessment as part of the TARGET integrated care program.

### 1.1. PCAM: Theoretical Foundation

The theoretical foundation of the PCAM builds on the INTERMED and Minnesota Complexity Assessment Method (MCAM), from which the tool originated [1,10,11,12,13]. As the MCAM was an American tool, it needed adaptations and new validation analyses in order to be applicable to a UK setting [1,2]. This led to the PCAM, which is an adapted version of the MCAM. The name of the tool was changed in order to move from a focus on ‘complexity’ to an emphasis on ‘patient centeredness’ [2]. Biopsychosocial complexity, described as “the interaction of biological (medical), psychological and social problems with a person’s health”, is a central theoretical concept within the PCAM [1]. In particular, Engel’s biopsychosocial model of illness supports the operationalization of the biological and psychological dimensions (i.e., the domain of ‘health and well-being’) as well as the social dimension (in the domain of ‘social environment’) of health and complexity [14]. As such, using the PCAM may help to deliver person-centered care, by taking the biopsychosocial needs, values, and preferences of individuals as starting point for collectively determining required referrals or other follow-up actions [15,16]. Although the evidence is (still) limited, person-centered care potentially improves quality of care and may lower work pressure in primary care when referrals following a PCAM assessment, for example to social care, are successful [17,18,19].

### 1.2. TARGET Program for Integrated, Person-Centered Care

We translated and psychometrically tested PCAM in the context of a recently developed Dutch-integrated, person-centered care program called TARGET [20]. TARGET is the acronym for ‘Targeting Advanced Resources in General practice to create Efficient, Tailored and holistic care for chronically ill patients’. This program was piloted from September 2020 to March 2021 in seven general practices located in the north of the Netherlands, where TARGET was developed. Data gathered during the pilot were used for psychometric assessment of the Dutch version of the PCAM. According to the Medical Research Committee Academic Hospital Maastricht/University Maastricht, the Netherlands, this pilot study was not prone to ethical review as the Dutch Medical Research (Human Subjects) Act (WMO) does not apply (#10117; 21 July 2020).

The development of TARGET was initiated by the primary care group ‘HZD’, located in a northern, predominantly rural area of the Netherlands. In brief, care groups support affiliated practices in organizing and delivering high-quality care to chronically ill patients; see Appendix A for more information about the role of care groups in the Netherlands and the Dutch primary care system in general. The TARGET program aims to facilitate care that is person-centered and delivered in an integrated way, thereby working towards better results in terms of the Quadruple Aim [21]. Although TARGET is intended for all chronically ill, the program was—for feasibility reasons—initially piloted among the subgroup with high care needs, as part of a larger-scale, ongoing realist evaluation. More information about how we selected high care need patients and the working mechanisms of TARGET can be found elsewhere [20,22].

PCAM was introduced in the TARGET program to facilitate a so-called person-centered needs assessment (hereafter referred to as ‘needs assessment’). This is a comprehensive conversation with a patient in general practice that takes about 30 to 45 min and is led by a trained care professional. Depending on the practice, this can be a general practitioner (GP) or practice nurse. The purpose is to discuss a patient’s biopsychosocial needs and subsequently use the PCAM’s action planning section to engage in shared decision-making about required follow-up actions. The PCAM served, for all seven practices, as a tool to measure the biopsychosocial needs as identified during the needs assessment, and make a shared decision about, and register, an action plan. A separate website was built to facilitate digital completion and retrieval of the PCAM. When professionals clicked on one of the answering options of the digital PCAM, the corresponding traffic light-color appeared. This website also provided a list of high care need patients which served to help professionals to select eligible patients for the needs assessment. For every high care need patient, a page with additional (visual) information about his/her care use during the previous year was available. Examples of provided information are the types of health problems for which a patient visited the primary care practice.

The ‘My Positive Health’ tool served as a primary conversation tool to support professionals and patients to engage in the needs assessment. This tool is derived from the ‘positive health’ concept as introduced by Huber and colleagues [23,24]. The main reason for choosing this instrument as the primary conversation tool was that most practices were familiar with the concept and some practices already had positive experiences with using the tool. Professionals received a needs assessment training in which they learned interview techniques inspired by ‘positive health’ and how to use the related tool during the needs assessment. The PCAM could be used by practices as a complementary conversation tool.

## 2. Materials and Methods

### 2.1. Translation and Contextualization

For the translation of the PCAM, guidelines as specified by the WHO were used [25]. Hence, our main goal was to reach cross-cultural and conceptual equivalence of the tool, rather than linguistic/literal equivalence. In agreement with WHO guidelines, a three-stage process was followed: (1) forward translation; (2) expert panel back-translation; and (3) pre-testing and cognitive interviewing [25].

In the first stage, author DH—whose mother tongue is Dutch, but is fluent in English—independently conducted a first forward translation of the PCAM into Dutch [25]. Authors RS and AE subsequently reviewed the translation and checked if any inadequate expressions were used or discrepancies existed between the translation and original PCAM. A bilingual expert panel was composed, consisting of the three authors (RS, DH, AE) involved in the forward translation, to reach consensus on a final forward translation of the PCAM. In stage 2, back-translation of the tool was conducted by an independent professional translator whose mother tongue is English and who was unfamiliar with the PCAM. Back-translation results were discussed and any identified discrepancies were resolved between the independent translator and authors RS, DH, and AE [25].

Finally, the translated tool was pre-tested with the target population (stage 3), i.e., Dutch primary care professionals involved in needs assessment as part of the TARGET pilot study. All professionals of the seven general practices were invited for an in-depth interview, organized per general practice, about the comprehensibility and contextual relevance of the translated PCAM. We developed a case description of a typical Dutch chronically ill patient with high care needs who is primarily monitored by the GP. The case description contained information about the patient’s biopsychosocial complexity, such as the number and type of conditions, latest blood values, housing circumstances, social network, and health literacy [26]. The Dutch case description was inspired by one of the patient cases offered by the University of Minnesota as training materials for PCAM users [26]. Before the interviews, respondents were asked to fill in the PCAM tool based on the provided case description. The interviews were either conducted individually by author DH, or by authors RS and DH collectively. Each interview started with general questions about how professionals experienced completing the PCAM (i.e., based on the case description) and what their impression of the tool was. After that, each individual PCAM item and corresponding answering categories were discussed, by asking professionals (1) to describe in their own words what the item addresses; (2) what answering category they chose; (3) how they chose their answer; and (4) if there were any unclear or contextually irrelevant words or phrases. The interviews were performed either digitally, using the ‘Zoom’ videoconferencing software, or via telephone. After finishing the interviews, authors RS, DH, and AE discussed the comments raised by the target population and composed a final version of the translated PCAM.

### 2.2. Psychometric Properties

#### 2.2.1. Population

Chronically ill with high care needs were included in the TARGET pilot study, and hence considered eligible for the needs assessment, if they were at least 18 years old and had sufficient mastery of the Dutch language. Patients who received palliative care and/or were institutionalized during the pilot study were excluded from the program. For psychometric testing, we used the PCAM results (i.e., 12 items, scored on a 4-point scale) of all patients who received a needs assessment during the pilot. From the electronic health record, we retrieved the following descriptive patient information: age, sex, weighted care utilization during the year preceding the needs assessment, number of chronic conditions, type of chronic conditions (only physical, only mental, combination of both), and prevalence of 28 common chronic conditions. All variables were measured at the time of the needs assessment. Because the robust weighted least squares (WLS) estimator is needed for a confirmatory factor analysis (CFA) of a tool with ordinal response categories like the PCAM, we considered, a priori, a minimum sample size of 200 patients for whom the PCAM was completed as sufficient [27,28,29].

#### 2.2.2. Analysis

Psychometric testing started with assessing the general properties, also called data quality, of the PCAM tool, as an indication of how well the translation and contextualization was performed. Hence, we computed frequency distributions of the answers to each PCAM item, to assess if the complete range of answering categories was used [30]. We assessed the number of missing values and calculated the median and mode score for each PCAM item. 

In terms of validity, the PCAM’s construct validity was assessed by performing a CFA. Factor analysis assumes that “measurable and observable variables can be reduced to fewer latent variables” [31]. Both of the previous factor structures as identified by Maxwell et al. [1] and Yoshida et al. [5] were tested in a CFA; see Figure 1 for a more detailed specification of the two tested structures and related PCAM items. Maxwell et al. [1] originally did not perceive the latter domain, then mentioned ‘action’, consisting of one item, as a separate and fourth construct. However, as this domain was further developed, called ‘service coordination’ and expanded with one item in the current version of the PCAM, we hypothesize that the latter domain constitutes an additional, fourth theoretical construct [2]. Hence, we tested a four-factor structure, consistent with the three-factor structure as identified by Maxwell et al. [1], but expanded with a separate factor for the latter domain. 

Due to the low ratio of items per factor of each structure, we handled missing data by listwise deletion. The CFA was conducted using the robust WLS estimator [28]. For each of the two factor structures, the following parameters were calculated and compared to assess what structure best fits the PCAM data. Factor loadings and standard errors of each PCAM item in relation to the assigned factor were derived from the CFA output. Loadings of at least 0.3 and 0.5 are generally considered acceptable and strong, respectively [5,32,33]. To assess model fit, it is recommended to use a variety of fit indicators that cover different aspects of model-data fit [27,34]. As a measure of global fit, we used the Standardized Root Mean Square Residual (SRMR) with a cut-off score of 0.08 or lower. To assess relative fit, i.e., fit of the tested models as compared to the unstructured model, the Tucker Lewis fit Index (TLI) was used. For the latter index, we considered a score of 0.90 or higher as an indication of acceptable model fit [27,34]. In addition to this, the Root Mean Square Error of Approximation (RMSEA) was calculated—using a cut-off score of 0.06 or lower [27].

In terms of reliability, we examined the internal consistency (i.e., degree to which items are intercorrelated) of the complete tool, and of items within each factor, for both factor structures [35]. A Cronbach’s alpha value of ≥0.70 and ≥0.80 are signs of acceptable and adequate internal consistency, respectively [35,36]. In the interpretation of the alpha value, we took into consideration that the alpha value of factors with a small (less than three) number of items may be reduced [35,36]. To assess the general properties, IBM SPSS Statistics (version 25) was used. For the psychometric tests, the statistical environment RStudio (version 1.4.1106) was used. 

### 2.3. Perceived Value, Feasibility and Face Validity

As part of the TARGET pilot, individual interviews with primary care professionals were organized at the end of the study period. The aim of the interviews was to obtain insight into the feasibility and acceptability of TARGET, including the perceived value, feasibility, and face validity of the PCAM in the context of the needs assessment. We aimed to interview 14 professionals in individual interviews, two of each of the seven practices participating in the TARGET pilot. A semi-structured interview guide was developed. The first interview was conducted by authors RS and DH collectively; subsequent interviews were conducted by either author RS or DH. The interviews were audio-taped and transcribed verbatim.

We used thematic analysis with an inductive approach for the qualitative data analysis [37]. A phased process was followed. In brief, authors RS and DH prepared the analyses by (re-)reading all transcripts, after which initial codes were applied. The first transcript was independently coded by the two researchers and the codes were compared and discussed for purposes of reflexivity. The remaining interviews were divided. Author RS drafted a first version of the subthemes and overarching themes that could be identified from the initial coding, which were discussed with DH. This helped to identify patterns of codes and relationships between the codes, which supported to understand, interpret, and report the main insights flowing from the data.

## 3. Results

### 3.1. Dutch Version of PCAM

During the forward translation, first adaptations were made to contextualize the PCAM. For instance, the term ‘client’ was replaced by ‘patient’, to adapt the PCAM for use in a primary care setting. Furthermore, as there was discussion about the interpretation of several words in the original version (e.g., ‘usual activities’ and ‘caregiving’), their meaning was verified with one of the developers of the PCAM to ensure correct translation. The back-translation showed small translation discrepancies that were resolved by discussion between authors RS, AE, and DH.

Twelve primary care professionals—i.e., six somatic practice nurses, two mental health practice nurses, one GP, one physician assistant, one nurse specialist, and one doctor’s assistant—pre-tested the translated PCAM. Most professionals reported that they considered the PCAM as a short but comprehensive tool to get a broad overview of the patient’s biopsychosocial situation. Challenges to complete the tool were also reported. For example, many professionals mentioned not being used to answering questions about a patient from their perspective as a professional. Hence, as they needed to complete the items based on their interpretation of a patient’s situation, professionals expected some degree of subjectivity, also between different professionals.

Discussion of each translated item and corresponding answering categories revealed that the Dutch translation was generally considered clear. Some words or phrases were perceived as complex and suggestions to rephrase were discussed. As an example, the literal Dutch translation of ‘inconsistency’ in the answering category “Safe, stable, but with some inconsistency” was changed into a simpler, but conceptually equivalent term. As the use of examples to clarify the content of each PCAM item was considered helpful, professionals often proposed to add new examples or adjust existing examples to optimize relevance for a Dutch context. Amongst others, we added participation in (community) associations as an example of the patients’ social network, as suggested by one professional. The final Dutch version of the PCAM can be found in Appendix B.

### 3.2. Study Participants

For 232 patients who received a needs assessment as part of the TARGET program, the PCAM was completed. The background characteristics of included patients are shown in Table 1. On average, patients were 72.5 years old and the majority was female (70.9%). More than eighty percent of patients had at least two chronic conditions. Although most (70.7%) patients had only physical condition(s), 27.6% of patients had a combination of physical and mental conditions and 1.8% had merely mental condition(s). During the year before the needs assessment, patients had a mean weighted care utilization of 46.9 contacts. Diabetes (55.1%), asthma (22.2%), and cancer (21.8%) were the top-three most prevalent chronic conditions.

### 3.3. PCAM General Properties

PCAM item response was high: 228 of the 232 PCAMs were completed without any missing values. In four PCAMs, there was one missing value (in items 7, 11, or 12); see Figure 2 for the frequency distribution and general properties of the 232 PCAM items scored using a four-point traffic light-style system indicating the growing need for action, ranging from ‘routine care’ (green) to ‘act now’ (red). For ten out of the 12 items, the most frequently used (i.e., mode) answer (in bold and delineated) was ‘routine care’, i.e., indicating the lowest need for action. The percentage of responses in ‘routine care’ ranged from 26% in item 2, related to the impact of physical problems on mental well-being, to 79% in item 8, related to financial resources. Two items (2 and 4), both concerning mental well-being, had a mode answer of ‘active monitoring’. ‘Routine care’ was also the median answer (indicated by the dotted line) for seven out of the 12 items, implying that this answer was scored for at least 50% of the patients in those items. The remaining five items, related to physical health needs, impact physical problems on mental well-being, (other concerns) mental well-being, social network, and need for other services, had a median of ‘active monitoring’.

‘Routine care’ (green) and ‘active monitoring’ (yellow) were the most frequent responses overall. On a patient level, 40% (*n* = 92) of patients scored only ‘routine care’ (green) or ‘active monitoring’ (yellow) in all PCAM items. Hence, the majority (60%; *n* = 140) of patients was indicated to need ‘plan action’ (orange) or ‘act now’ (red) on at least one of the 12 PCAM items. Of those 140 patients, 84% (*n* = 117) was indicated to need ‘plan action’ or ‘act now’ in at least one item of the domain of ‘health and well-being’. The remaining 16% (*n* = 23) did not score the orange or red option in the first domain, whereas they did need ‘plan action’ or ‘act now’ in at least one item of the remaining three domains.

### 3.4. Psychometric Properties

The CFA was conducted with the 228 complete PCAMs. In Table 2, the factor loadings of the two assessed structures are provided. All loadings are above the minimally acceptable threshold of 0.3. The majority of loadings are above 0.5, indicating that most loadings can be classified as strong. Exceptions (in bold) are, for factor structure 1, the loading of item 1 (physical health needs) on factor 1 (health and well-being), and the loading of item 8 (financial resources) on factor 2 (social environment). For factor structure 2, item 1 (physical health needs) and item 8 (financial resources) also showed acceptable but not strong loadings on factor 2: medicine-oriented complexity.

Table 3 shows the goodness-of-fit indices and Cronbach’s alpha values that were calculated for the two structures. For structure 1, all indices (i.e., SRMR, TLI, and RMSEA) met the thresholds of acceptable fit. For structure 2, none of the indices met the thresholds of acceptable fit. The Cronbach’s alpha of the complete PCAM tool (0.83) met the threshold of 0.8, indicating adequate internal consistency. For factor structure 1 [1], the Cronbach’s alpha values of factors 3 and 4 met the threshold of acceptable internal consistency (i.e., 0.70), whereas the values for factors 1 and 2 were just below the threshold, with values of 0.69 and 0.66. If item 1, concerning physical health needs, was dropped from factor 1, the Cronbach’s alpha value of factor 1 would increase from 0.69 to 0.72. If item 8, concerning financial resources, was dropped from factor 2, the Cronbach’s alpha value of factor 2 would increase from 0.66 to 0.70. With regards to the second tested factor structure [5], only the first factor showed adequate internal consistency (0.8), whereas the Cronbach’s alpha value of the second factor (0.59) was below the threshold. Again, if item 1 was dropped, the Cronbach’s alpha value of factor 2 would be 0.64, and if item 8 was dropped, the current Cronbach’s alpha value (0.59) would be maintained. No other items would lead to improved Cronbach’s alpha values if dropped.

### 3.5. Perceived Value, Feasibility and Face Validity

As intended, we interviewed two professionals of each of the seven practices, except for one practice for which we interviewed three professionals. Hence, 15 professionals were interviewed in total. All interviews were performed individually, except for one interview with two professionals. Amongst the participants, there were six GPs, five somatic practice nurses, two mental health practice nurses, one physician assistant, and one doctor’s assistant. Twelve of the 15 participants were female. Their mean age was 50 years (SD = 12.5). The youngest was 22 and the oldest 63 years old. On average, they had 14 years of work experience in primary care (SD = 9.6). Below, the value of the PCAM is described by the different functions the tool had in the current study (i.e., to facilitate measurement and action planning, and serve as a complementary conversation tool).

#### 3.5.1. PCAM as Measurement Tool

Professionals reported that they mainly perceived the PCAM as a measurement tool. It enabled measurement of the outcomes of the needs assessments and helped some professionals to determine how complete their ‘picture’ of a patient is. Nevertheless, professionals saw the measurement function of the PCAM as predominantly valuable for scientific research and as less important for daily practice:
*“We have got more measurement tools, for instance for people with COPD. It could have some value [to use the PCAM as measurement tool], but on the other hand I think: we have got so many measurement tools. With the conversation [the needs assessment] you mainly focus on: Who is in front of you? What can you do for someone?”*(Primary care professional 2)

Furthermore, some professionals reported a disagreement between the PCAM (as a measurement tool) and the needs assessment: although a professional interpretation of a patient’s situation is needed to fill in the PCAM, the needs assessment should be focused on the patient perspective:
*“Such a conversation [the needs assessment] is about things that are very important for the patient. […] So it happens that topics are not addressed which I, as a caregiver, wanted to address but the patients did not want to. And when you then fill in the PCAM, you sometimes miss information. So it is a matter of translating the thoughts of the patient to how the professional interprets it.”*(Primary care professional 3)

Most needs assessments were conducted by practices nurses, who subsequently completed the PCAM together with a GP. The GP helped to fill in the PCAM—based on prior experiences with the patient instead of the needs assessment—because practice nurses were sometimes unsure whether they interpreted the situation of the patient correctly and in line with the interpretation of the GP. Some professionals reported that their interpretations often matched, whereas others indicated that objective completion of the PCAM was difficult, as assessments of the complexity of a patient’s needs can differ between professionals. Completing the PCAM together was valued by professionals. It offered the chance to share new information of the patient that was discussed during the needs assessment, and to collectively think about the required actions for a patient. Practice nurses also saw it as a way to create shared responsibility with the GP to act upon the action plans.

#### 3.5.2. PCAM as Action Planning Tool

Many professionals considered the action planning section of the PCAM as clear and helpful to determine and register follow-up actions. It stimulated critical thinking about the needed follow-up actions after the needs assessment:
*“It is helpful to have a sort of evaluation moment at the end of such a conversation [the needs assessment]. […] I like to wrap it up like: What types of challenges does the patient encounter? And what is already going well? The PCAM is suited for this, in my opinion.”*(Primary care professional 9)

Others argued that the existing electronic health record, in addition to the ‘My Positive Health’ primary conversation tool, already facilitates action planning sufficiently, so the PCAM is redundant for this purpose. In addition, professionals indicated that the action planning section was not always completed, as the situation of the patient did not call for (new) follow-up actions or because professionals were unsure about how to fill in this section. It was therefore suggested to practice the completion of the action planning section with colleagues before starting to use it.

#### 3.5.3. PCAM as Conversation Tool

A small number of professionals used the PCAM as a second, complementary conversation tool next to ‘My Positive Health’. They argued that it helped them to adequately prepare and conduct the needs assessment. Some PCAM questions (e.g., about alcohol use and debts) were not included in ‘My Positive Health’, but were seen as important and complementary questions to address during the needs assessment. Hence, the PCAM helped to get a “complete picture” of a patient. However, some professionals mentioned they perceived those questions as emotionally charged and were therefore challenging to ask.

Most professionals did not use the PCAM as a complementary conversation tool and considered ‘My Positive Health’ as sufficient for this purpose. Some professionals mentioned specific shortcomings of the PCAM as a conversation tool. First, the PCAM does not have a patient version, which limits the opportunity for patients to prepare the needs assessment. Furthermore, professionals indicated that the PCAM is mainly focused on determining actions, rather than on the needs assessment itself. For instance, the PCAM does not facilitate summarizing what was discussed about the situation of a patient. Only the registration of actions is supported.

#### 3.5.4. Feasibility

In terms of feasibility, most professionals perceived the PCAM as a clear and easy to use tool. It only took them a few minutes to fill in the items, which was most frequently done right after the needs assessment. However, some professionals did find the PCAM to be time-consuming and therefore did not always manage to fill in the PCAM items shortly after the needs assessment. To make it less time-consuming and improve efficiency, professionals indicated the need to integrate the PCAM into the electronic health record instead of having to access the tool via a separate website. This would also help to obtain a comprehensive overview of the needs assessment and related actions, as all information is stored in one location.

#### 3.5.5. Face Validity

Professionals indicated that the PCAM contains legitimate questions for a holistic, biopsychosocial conversation with a patient. However, professionals also mentioned validity concerns of the tool. Firstly, the differences between the answering options were seen as large. Hence, professionals were sometimes unable to find the correct answer for the specific situation of the patient. To overcome this, they suggested to create an open field to add some more detailed information about the patient. Moreover, some questions and answering options were considered complex, asking for two assessments at once. For instance, the answering option ‘financially insecure, very few resources, immediate challenges’ contains an assessment of the patient’s financial situation (‘financially insecure, very few resources’) and an assessment of the urgency to respond to the situation (‘immediate challenges’). This also shows the assumption underlying many PCAM items that a more complex situation asks for a higher level of intervention, which is sometimes incorrect:
*“A red score on ‘financial problems’ does not have to indicate that there is a problem. We’ve got one patient who scores definitely ‘red’ in terms of the financial situation, but she still manages it with some help. So it is not really a problem, but I still have to score it as a problem. […] It should be a green score, but that is not possible because green says there are no financial problems.”*(Primary care professional 8)

The needs assessments were only done with patients with high care needs, but professionals reported that they rarely indicated urgent needs for intervention with the PCAM. Professionals still valued a holistic conversation with these patients.

## 4. Discussion

This study aimed to create a contextualized Dutch version of the PCAM, increase insight into the psychometric properties of the tool, and test the perceived value, feasibility, and face validity of the PCAM as a measurement, action planning, and (complementary) conversation tool. The results show ambiguity, particularly across the quantitative and qualitative analyses. The internal consistency of the complete tool was of an adequate level (Cronbach’s alpha is 0.83). In terms of construct validity, the CFA confirmed that the four-factor structure of Maxwell et al. [1] fitted the PCAM data well, in contrast to the two-factor structure of Yoshida et al. [5]. However, the qualitative results revealed that the PCAM needs some validity improvements. Nonetheless, professionals also indicated that the PCAM has value for measurement, as a first function. In terms of the other functions of the PCAM, the tool was mainly valued for action planning, and was only used by a minority of professionals as complementary conversation tool.

Despite concerns about the face validity of the tool, the quantitative results confirm that the PCAM is adequate for its first function in the current study, i.e., to support measurement of needs assessment outcomes. Similar to previous studies, the internal consistency was of an adequate level [1,5]. All 12 PCAM items contribute to valid and reliable measurement of the related construct, except for item 1 and item 8 in both tested factor structures. For item 1, this may be explained by the content of the item: item 1 is purely focused on physical health needs whereas items 2 to 4 (amongst others) focus on mental well-being. With regards to item 8 about financial resources, the percentage of patients who were indicated to only need routine care was substantially higher (i.e., 79%) than in the other three items of the domain ‘social environment’ (i.e., between 42 and 60%). There are several possible explanations for this. First, as this study was conducted in a predominantly rural area of the Netherlands, with less deprivation than in other, more urban regions of the country, the prevalence of financial issues may actually be lower [38]. However, previous research shows that the target population of this study, i.e., high care need patients, more often has financial problems than found in this study [39,40,41,42]. Therefore, a second possible explanation is that financial problems were not always identified and acted upon. In line with this study, research shows that barriers (e.g., taboos) exist to discuss financial issues in primary care [43,44,45]. In a recent study on the Japanese version of the PCAM, item 8 was also identified as problematic for the validity of the tool [46]. However, it was still considered an important topic to address in primary care, in line with the findings of the current study [46].

A point of criticism regarding the PCAM as measurement tool, expressed by the interviewed professionals, is that a professional interpretation of the patient’s situation is required to fill in the PCAM, whereas the needs assessment should be focused on the patient perspective. This is quite remarkable as professional interpretation is inherent to each medical profession and does not necessarily mean the patient perspective is overlooked. Furthermore, the PCAM requires a focus on the patient experiences during the assessment in order to adequately fill in the 12 items. Nonetheless, this point of criticism uncovers a difference in the theoretical models underpinning the PCAM and the needs assessment. The adequate fit of the four-factor model of Maxwell et al. [1], in which the biological-psychological domains (combined into ‘health and well-being’) and social domain (‘social environment’) were identified as two of the four separate constructs, shows that the biopsychosocial model has informed the PCAM. Although the biopsychosocial model does consider multiple aspects of the person, it does not have an explicit focus on the individual personhood of patients, i.e., how patients perceive their situation [47]. However, this is a crucial element of the more comprehensive and contemporary concepts of person-centered holistic care, described as the aims of the needs assessment [15,47,48]. To make the PCAM more compatible with the needs assessment, creating a patient (next to a professional) measurement tool may be helpful. It should, however, be noted that directly measuring the subjective experiences of patients in a valid and reliable way is challenging, as demonstrated by the efforts to transform the patient-directed conversation tool ‘My Positive Health’ into a comprehensive measurement instrument [49].

The second function of the PCAM, i.e., to support action planning, was appreciated by many professionals in the current study. This corresponds with previous studies, describing that the action component of the PCAM was, in particular, helpful to guide patients towards the right intervention, referral, or other follow-up action [1,2,7]. Doing so has the potential to lower the work pressure in primary care and ensure patients receive the care or support best fitting their needs [1,2,7]. As shown in this study, it may be helpful to complete the action plan with a team of professionals after it was discussed with the patient. In particular, patients with complex biopsychosocial issues—for whom the ‘regular’ care paths are often insufficient—may benefit from the collective expertise of various professionals [50,51,52]. Yet, the number of identified actions with the label ‘plan action’ or ‘act now’ in one of the PCAM items was lower (i.e., for 60% of patients) than expected in a population with high care needs [50,51,52]. In addition, professionals had mainly identified those actions in ‘health and well-being’, a domain that is traditionally addressed by primary care. For a minority of patients (i.e., 16%), the actions were indicated to only relate to the other three, more social domains. An explanation may be that professionals have more knowledge, skills, and trust to discuss items and plan actions closely related to the domain of primary care than those related to the more social domains [44,45]. Furthermore, to succeed in social actions or referrals to other settings and professionals, strong network relations are crucial, but may not be present or developed sufficiently during the course of the pilot study [44,45,53].

The use and appreciation of the PCAM in its third function, i.e., as a conversation tool, was limited. Professionals reported several reasons for this; for example, the fact that the PCAM lacks a patient version which prepares patients for the needs assessment. This finding conflicts with previous PCAM studies, in which the tool was valued as a framework to guide the conversation and considered a helpful instrument to improve the quality and openness of communication [1,2,7].However, it should also be noted that the PCAM was not fully tested as a conversation tool in this study. This is due to the fact that some professionals already had positive experiences with the primary conversation tool ‘My Positive Health’ before the start of the pilot. This may have served as a barrier to using a second and new tool. The needs assessment training, which was largely influenced by ‘positive health’, possibly also ‘steered’ professionals towards using ‘My Positive Health’.

### 4.1. Practical Implications, Future Research and Policy

The face validity concerns as expressed in this study, in addition to the finding that the current concept of ‘person-centered care’ is not fully supported by the PCAM, call for a revision of the tool as both a measurement and conversation tool. In addition, a patient version of the tool is needed, as was also suggested by the developers of the PCAM [1,54]. This can help provide more explicit attention for the ‘individual personhood’ of a patient. To make the tools relevant and appealing, an expert group with patients should be formed. Patients with different characteristics, for instance in terms of age, socioeconomic status, and health literacy, should be included in this expert group to ensure the tool is relevant for the diverse population of patients who (often) consult primary care. For the revision of the professional version, a professional expert group is helpful. In this process, it is important to ensure that the good properties of the tool are maintained: the adequate psychometric qualities and the action planning component. In terms of policy, the findings of the current study point to the importance of a well-functioning network surrounding the primary care practice. When there are strong connections with, for instance, the social domain, professionals may have more confidence to act upon issues of a social kind when these are identified.

### 4.2. Strengths and Limitations

A strength of the current study is the mixed-methods design. This helped to compare the quantitative insights into the PCAM’s measurement qualities with the experienced validity and value. As such, a rich understanding of the value of the PCAM in its different functions was obtained. A limitation is that some psychometric properties of the tool, i.e., the stability, inter-rater reliability, and the criterion validity, were not studied. The main reason for this was that the pilot study was aimed at investigating the feasibility and acceptability of the TARGET program. It was therefore considered out of the scope of the study to, for instance, ask professionals to complete a second measurement instrument next to the PCAM (to study criterion validity). Previous studies have investigated criterion validity, but the results are mixed [1,5,46]. As far as we are aware, the other two psychometric properties have not yet been studied. However, it is arguable whether a high inter-rater reliability is attainable. Professionals reported that some degree of subjectivity is inherent to the professional interpretation of a patient’s situation.

## 5. Conclusions

The PCAM is an adequate biopsychosocial measurement tool. Furthermore, it helps professionals—when the professionals have strong connections with their network and referral options—to plan actions based on a needs assessment with (high care need) patients in primary care. However, to support a holistic, person-centered needs assessment, the tool needs a patient version and revision—while keeping the strong elements—to fully meet the features of person-centered care as the concept is described today.

## Figures and Tables

**Figure 1 ijerph-18-11785-f001:**
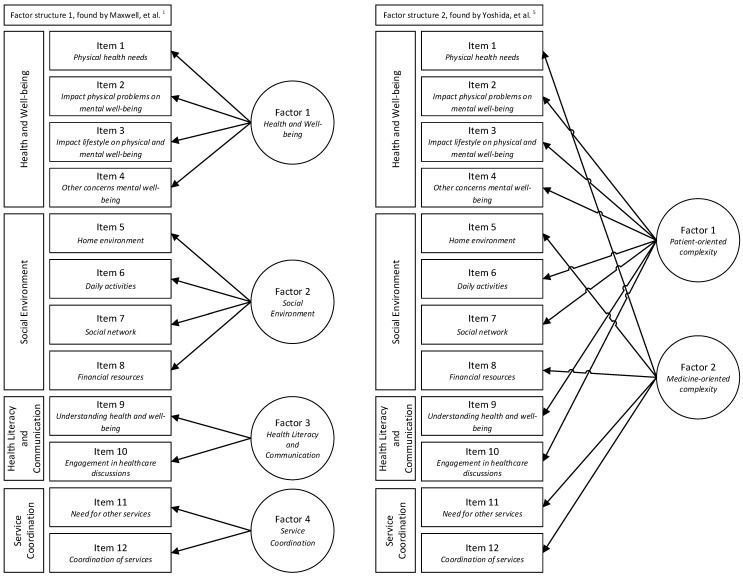
Overview of the two different factor structures as identified for the PCAM.

**Figure 2 ijerph-18-11785-f002:**
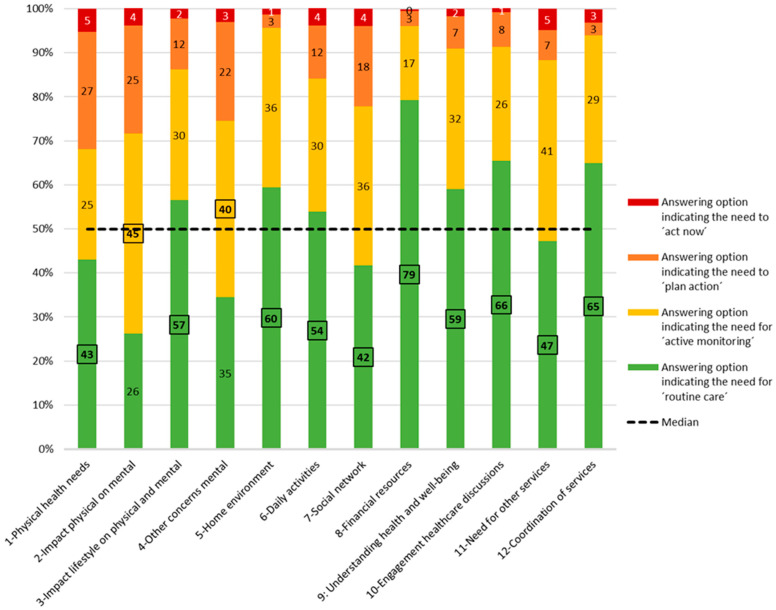
PCAM general properties.

**Table 1 ijerph-18-11785-t001:** Background characteristics of study participants (*n* = 232).

	N (SD/Percentage)
Age in years ^a^, mean (SD)	72.5 (±14.1)
Age in years ^a^, number (percentage)	
<65 years	55 (23.7%)
≥65 and <80 years	85 (36.6%)
≥80 years	92 (39.7%)
Sex, number (percentage)	
Male	64 (29.1%)
Female	156 (70.9%)
Weighted care utilization, mean (SD) ^b^	46.9 (±20.4)
Number of chronic conditions ^c^, number (percentage)	
One	41 (18.2%)
Two	78 (34.7%)
Three of more	106 (47.1%)
Type of chronic condition(s) ^c^, number (percentage)	
Only physical	159 (70.7%)
Only mental	4 (1.8%)
Combination of physical and mental	62 (27.6%)
Chronic conditions ^c^, number (percentage)	
Diabetes mellitus	124 (55.1%)
Asthma	50 (22.2%)
Cancer	49 (21.8%)
Chronic obstructive pulmonary disease (COPD)	45 (20.0%)
Coronary heart diseases	37 (16.4%)
Chronic back or neck disorder	35 (15.6%)
Heart failure	33 (14.7%)
Mood disorders	32 (14.2%)
Heart arrhythmia	31 (13.8%)
Osteoarthritis	31 (13.8%)
Visual disorders	23 (10.2%)
Stroke (including TIA)	21 (9.3%)
Anxiety disorders	19 (8.4%)
Burnout	12 (5.3%)
Osteoporosis	10 (4.4%)
Rheumatoid arthritis	8 (3.6%)
Dementia including Alzheimer’s	7 (3.1%)
Hearing disorders	7 (3.1%)
Endocardial conditions, valvular conditions	5 (2.2%)
Chronic alcohol abuse	4 (1.8%)
Mental retardation	3 (1.3%)
Migraine	3 (1.3%)
Epilepsy	1 (0.4%)
Parkinson’s disease	1 (0.4%)
Personality disorders	1 (0.4%)
Schizophrenia	1 (0.4%)

Note: The characteristics of age and weighted care utilization had no missing values; the remaining characteristics had either 7 (3%) missing values (i.e., number, type, and prevalence of chronic conditions) or 12 (5%) missing values (i.e., sex). ^a^ Measured at the time of the needs assessment. ^b^ Based on the care use during the year before the needs assessment and weighted for the intensity of types of consultations used; applied weights are described elsewhere [20,21]. ^c^ Based on the care use for chronic conditions during the one and a half year period preceding the needs assessment. The conditions of congenital cardiovascular anomaly and HIV/AIDS were not included in the table as their prevalence was zero.

**Table 2 ijerph-18-11785-t002:** Factor loadings of items within the two assessed factor structures.

Factor Structure 1, by Maxwell et al. [1]		Factor Structure 2, by Yoshida et al. [5]	
Factors	Factor Loadings	Factors	Factor Loadings
Factor 1: health and well-being	**Item 1: 0.432** Item 2: 0.692 Item 3: 0.630 Item 4: 0.897	Factor 1: patient-oriented complexity	Item 2: 0.654 Item 3: 0.596 Item 4: 0.832 Item 6: 0.684 Item 7: 0.713 Item 9: 0.701 Item 10: 0.701
Factor 2: social environment	Item 5: 0.681 Item 6: 0.748 Item 7: 0.783 **Item 8: 0.409**	Factor 2: medicine-oriented complexity	**Item 1: 0.426** Item 5: 0.664 **Item 8: 0.382** Item 11: 0.773 Item 12: 0.842
Factor 3: health literacy and communication	Item 9: 0.860 Item 10: 0.853		
Factor 4: service coordination	Item 11: 0.827 Item 12: 0.917		

The factor loadings in bold are below the threshold of 0.5 indicating they are acceptable but not strong.

**Table 3 ijerph-18-11785-t003:** Fit indices and Cronbach’s alpha values for the two assessed factor structures.

	Factor Structure 1, Found by Maxwell et al. [1]	Factor Structure 2, Found by Yoshida et al. [5]
SRMR ^a^	0.061 *	0.098
TLI ^b^	0.968 *	0.885
RMSEA ^c^	0.057 *	0.109
Cronbach’s alpha	Factor 1: 0.69 Factor 2: 0.66 Factor 3: 0.75 * Factor 4: 0.75 *	Factor 1: 0.8 * Factor 2: 0.59

^a^ SRMR is the Standardized Root Mean Square Residual, acceptable fit ≤ 0.08. ^b^ TLI is the Tucker Lewis fit Index, acceptable fit ≥ 0.90. ^c^ RMSEA is the Root Mean Square Error of Approximation, acceptable fit ≤ 0.06. * Fit indices and Cronbach’s alpha values that meet the thresholds indicating acceptable fit and internal consistency, respectively.

## Data Availability

We used data from general practices connected to primary care group ‘HZD’ in this study. As the data is owned by the general practices and ‘HZD’, requests for access to this data should be submitted to ‘HZD’ (info@hzd.nu).

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
