# Peer review of "The Patient Centered Assessment Method (PCAM) for Action-Based Biopsychosocial Evaluation of Patient Needs: Validation and Perceived Value of the Dutch Translation"

_ijerph, 2021, doi:10.3390/ijerph182211785_

Round 1

Reviewer 1 Report

It is very interesting topic.

  1. Abstract - it doesn't meet the publisher requirements as it should contain 200 words, while the present one contains more then 200 words.
  2. Abstract contains a lot of info on the results but it is difficult to figure out what is the novelty of these results or what is their added value. 
  3. It is well systematized and planned research. 
  4. However the research focus on the certain type of population rather old-age population - therefore some discussion should also apply to the application of such tool in the context of population with the different characteristics. 

I would suggest the correction of abstract and discussion part (points 1,2,4)

Author Response

Dear Professor Tchounwou,                                     Maastricht, November 5, 2021

We are very grateful to be offered the opportunity to revise and resubmit our manuscript entitled: “The Patient Centered Assessment Method (PCAM) for action-based biopsychosocial evaluation of patient needs: validation and perceived value of the Dutch translation” for publication in the special issue about "The Importance of Person-Centered Primary Care" of The International Journal of Environmental Research and Public Health.

We appreciate the helpful feedback that was provided and believe that these comments were relevant and useful to improve our manuscript. Please find below our responses to the comments and reference to the changes we made in the manuscript. We have added our revised manuscript with the changes marked up using the “Track Changes” function. Please also find our revised manuscript with the changes accepted. The latter document also contains the right references, as we were unable to change these in the first document.

We sincerely think that we have improved the quality of our paper and we hope that you will consider it for publication in The International Journal of Environmental Research and Public Health.

With kind regards, also on behalf of the co-authors,

Rowan Smeets, MSc

Maastricht University
Faculty of Health, Medicine and Life Sciences
Care and Public Health Research Institute (CAPHRI)
Department of Health Services Research
P.O. Box 616
6200 MD Maastricht, the Netherlands
T: +31(0)43 3881711 / F: +31(0)43 3884162
[email protected]

Comments and Suggestions for Authors from Reviewer 1

Comment #1: It is very interesting topic.
Our response: We thank the reviewer for the compliment.

Comment #2: Abstract - it doesn't meet the publisher requirements as it should contain 200 words, while the present one contains more then 200 words.
Our response: Thank you for your alertness. We counted the number of words in the abstract again, but it contains only 196 words. We adapted the abstract based on further comments from the reviewers: in so doing, we made sure not to exceed the 200 word limit.

Comment #3: Abstract contains a lot of info on the results but it is difficult to figure out what is the novelty of these results or what is their added value. 
Our response: We agree with the reviewer and have added some more information to the revised manuscript, at the end of the abstract, about why the results are novel and of added value. See lines 27-29 on the first page: These rich, mixed-method insights can help to improve the value of the PCAM, as one of the few multifunctional tools to support professionals in holistic assessments.” To create room for adding this information, we shortened other parts of the abstract a bit, without losing important information to understand the current study.

Comment #4: It is well systematized and planned research. 
Our response: We thank the reviewer for the compliment.

Comment #5: However the research focus on the certain type of population rather old-age population - therefore some discussion should also apply to the application of such tool in the context of population with the different characteristics. 
Our response: With approximately 75% of our participants over the age of 65, we agree that our included patient population is relatively old-age. However, taking a look at the literature about the characteristics of patients who most often visit the general practice, we believe it is quite a representative population for primary care. That being said, we acknowledge the importance of taking into account a variety of patient characteristics, especially when designing a patient version of the PCAM. Therefore, in response to this reviewer comment, we have added the following to the discussion section: Patients with different characteristics, for instance in terms of age, socioeconomic status and health literacy, should be included in this expert group to ensure the tool is relevant for the diverse population of patients who (often) consult primary care.” (page 11, lines 669-671).

Reviewer 2 Report

  1. After the introduction, it is crucial that you include sections that help to describe the PCAM tool. What is this tool? Who developed it etc.  This information might be under the materials and methods section but it is beneficial to put them after the introduction for clarity.
  2. I also suggest that the authors include a brief section for empirical and theoretical literature review. This section is critical to position the study.
  3. This section (TARGET program) should be moved from the material and methods. I propose that you put it before this section under the introduction. Putting it under material and methods will confuse the readers
  4. Section 2.2. Translation and contextualization has a lot of missing citations
  5. Authors should separate the methodology used in the current study and the methodology used when the tool was developed. Describing how the original tool was developed should act as the literature review. The methodology section should describe the method used in the current study.
  6. Authors should compare the results with those of other studies to ensure that there is a discussion.
  7. The conclusion should be improved and include policy recommendations if they are available.

Author Response

Dear Professor Tchounwou,                                     Maastricht, November 5, 2021

We are very grateful to be offered the opportunity to revise and resubmit our manuscript entitled: “The Patient Centered Assessment Method (PCAM) for action-based biopsychosocial evaluation of patient needs: validation and perceived value of the Dutch translation” for publication in the special issue about "The Importance of Person-Centered Primary Care" of The International Journal of Environmental Research and Public Health.

We appreciate the helpful feedback that was provided and believe that these comments were relevant and useful to improve our manuscript. Please find below our responses to the comments and reference to the changes we made in the manuscript. We have added our revised manuscript with the changes marked up using the “Track Changes” function. Please also find our revised manuscript with the changes accepted. The latter document also contains the right references, as we were unable to change these in the first document.

We sincerely think that we have improved the quality of our paper and we hope that you will consider it for publication in The International Journal of Environmental Research and Public Health.

With kind regards, also on behalf of the co-authors,

Rowan Smeets, MSc

Maastricht University
Faculty of Health, Medicine and Life Sciences
Care and Public Health Research Institute (CAPHRI)
Department of Health Services Research
P.O. Box 616
6200 MD Maastricht, the Netherlands
T: +31(0)43 3881711 / F: +31(0)43 3884162
[email protected]

Comments and Suggestions for Authors from Reviewer 2

Comment #1: After the introduction, it is crucial that you include sections that help to describe the PCAM tool. What is this tool? Who developed it etc.  This information might be under the materials and methods section but it is beneficial to put them after the introduction for clarity.
Our response: We agree that it is helpful for the reader to find information about the PCAM in a separate paragraph after the introduction. The second paragraph of the former version of the introduction already contained information about the theoretical background and development process of the PCAM. Therefore, we have used and elaborated this second paragraph as a separate PCAM paragraph after the introduction (see paragraph 1.1. PCAM: theoretical foundation, pages 2-3, lines 88-104). This implies that the explanation of how the PCAM relates to person-centered care was now also moved to paragraph 1.1. To make this relation explicit at the beginning of the introduction, we have added the following words to the beginning of the introduction: “Hence while the PCAM is primarily a conversation tool to take a comprehensive, person-centred approach to patients, it also supports measurement and monitoring of patient needs [1].” (page 2, line 42).

As we had to change the order of the text, the references also needed to be updated. In the document with the ‘Track changes’, we were unable to change the references as they were not connected to our citation manager any more. Therefore, we have also added our revised manuscript with all changes accepted and the right references.

Comment #2: I also suggest that the authors include a brief section for empirical and theoretical literature review. This section is critical to position the study.
Our response: We have added a separate paragraph about how the PCAM was (theoretically) developed after the introduction of the revised manuscript (see our response to comment #1). The empirical literature review, providing current insights into the psychometric properties of the PCAM, was already provided in the third paragraph of the former version of the introduction. However, we have elaborated this part with information about the internal consistency of the tool found in previous studies (page 2, lines 66-69): “While existing studies conclude that the PCAM has good internal consistency, the insights into the theoretical constructs (also described as ‘factors’) measured by the tool are conflicting.”

Comment #3: This section (TARGET program) should be moved from the material and methods. I propose that you put it before this section under the introduction. Putting it under material and methods will confuse the readers
Our response: We understand that this might confuse readers. As the reviewer suggested, we have created a separate paragraph with information about TARGET after the introduction (see paragraph 1.2. TARGET program, pages 3-4, lines 105-152).

Comment #4: Section 2.2. Translation and contextualization has a lot of missing citations
Our response: We have added citations to this paragraph in the revised manuscript, in particular to the WHO guidelines we used during this process.

Comment #5: Authors should separate the methodology used in the current study and the methodology used when the tool was developed. Describing how the original tool was developed should act as the literature review. The methodology section should describe the method used in the current study.
Our response: One of the aims of this study was to translate the original version of the PCAM to Dutch. Hence, the translation and contextualization process of the Dutch version of the PCAM is part of the methodology used in the current study and was thus described in the methods section. This is in line with how comparable studies combining translation and validation methodologies are written.

Comment #6: Authors should compare the results with those of other studies to ensure that there is a discussion.
Our response: We agree that additional comparison with previous studies would improve the quality of the discussion. Therefore, we compare our findings about internal consistency with previous studies in the revised manuscript: “Similar to previous studies, the internal consistency was of an adequate level [1,16].” (page 9, lines 589-590). In addition, when discussing that the PCAM was to a limited extent appreciated as conversation tool, we have added the following to the revised manuscript: “This finding conflicts with previous PCAM studies, in which the tool was valued as a framework to guide the conversation and considered a helpful instrument to improve the quality and openness of communication.” (page 10, lines 653-655).

Comment #7: The conclusion should be improved and include policy recommendations if they are available.
Our response: We now realize that implications/recommendations for policy were missing in the previous version of our manuscript. Therefore, we have added the following to paragraph 4.1., which we named ‘Practical implications, future research and policy’ (page 11, lines 674-768) to the revised manuscript: “In terms of policy, the findings of the current study point to the importance of a well-functioning network surrounding the primary care practice. When there are strong connections with, for instance, the social domain, professionals may have more confidence to act upon issues of a social kind when these are identified.” In addition, we added a short reference to this policy recommendation in the conclusion: “Furthermore, it helps professionals to – when professionals have strong connections with their network and referral options – plan actions based on a needs assessment with (high care need) patients in primary care.” (page 11, lines 695-696)